# DNA methylome regulates virulence and metabolism in *Pseudomonas syringae*

Jiadai Huang[1,2,3,4†], Fang Chen[1†], Beifang Lu[1], Yue Sun[1], Youyue Li[1], Canfeng Hua[1], Xin Deng[1,2,3,4,5]*

[1]Department of Biomedical Sciences, City University of Hong Kong, Hong Kong, China; [2]Shenzhen Research Institute, City University of Hong Kong, Shenzhen, Guangdong, China; [3]Chengdu Research Institute, City University of Hong Kong, Chengdu, China; [4]Institute of Digital Medicine, City University of Hong Kong, Hong Kong, China; [5]Tung Research Centre, City University of Hong Kong, Hong Kong, China

*For correspondence:
xindeng@cityu.edu.hk

[†]These authors contributed equally to this work

## eLife Assessment

This **valuable** study presents findings on DNA methylation as an efficient epigenetic transcriptional regulating strategy in bacteria. The authors utilized single-molecule real-time sequencing to profile the DNA methylation landscape across three model pathovars of *Pseudomonas syringae*, identifying significant epigenetic mechanisms through the Type-I restriction-modification system, which includes a conserved sequence motif associated with N6-methyladenine. The evidence presented is **solid** and the study provides novel insights into the epigenetic mechanisms of *P. syringae*, expanding the understanding of bacterial pathogenicity and adaptation.

**Abstract** Bacterial pathogens employ epigenetic mechanisms, including DNA methylation, to adapt to environmental changes, and these mechanisms play important roles in various biological processes. *Pseudomonas syringae* is a model phytopathogenic bacterium, but its methylome is less well known than that of other species. In this study, we conducted single-molecule real-time sequencing to profile the DNA methylation landscape in three model pathovars of *P. syringae*. We identified one Type I restriction–modification system (HsdMSR), including the conserved sequence motif associated with $N^6$-methyladenine (6mA). About 25–40% of the genes involved in DNA methylation were conserved in two or more of the strains, revealing the functional conservation of methylation in *P. syringae*. Subsequent transcriptomic analysis highlighted the involvement of HsdMSR in virulent and metabolic pathways, including the Type III secretion system, biofilm formation, and translational efficiency. The regulatory effect of HsdMSR on transcription was dependent on both strands being fully 6mA methylated. Overall, this work illustrated the methylation profile in *P. syringae* and the critical involvement of DNA methylation in regulating virulence and metabolism. Thus, this work contributes to a deeper understanding of epigenetic transcriptional control in *P. syringae* and related bacteria.

## Introduction

*Pseudomonas syringae* inflicts leaf spots and cankers on plants globally, and its adequate control poses significant challenges. More than 60 pathovars have been identified, which infect almost all economically important crops in the world (**Bull et al., 2010**; **Xin et al., 2018**). *P. syringae* pv. *phaseolicola* 1448A (Psph), one of the classical strains of *P. syringae*, can cause severe halo blight of common beans (*Phaseolus vulgaris*), which raises it from a common pathogen to a molecular plant–pathogen

bacterium (*Arnold et al., 2011*). *P. syringae* pv. *tomato* DC3000 (*Pst*) and *P. syringae* pv. *syringae* B728a (*Pss*) are two other model strains whose natural host plants are tomatoes and beans, respectively (*Xin and He, 2013*; *Monier and Lindow, 2005*). The most important weapon of *P. syringae*, and the first to be characterized, is the Type III secretion system (T3SS) encoded by *hrp* and *hrc* gene clusters, which are flanked by the conserved effector locus and deliver T3 effectors into host cells (*Clarke et al., 2010*).

The expression of T3SS genes is inhibited in nutrient media such as King's B (KB), whereas it is induced in minimal medium or plant cells (*Rahme et al., 1992*; *Xiao et al., 2004*; *Xiao et al., 1992*). The HrpRSL pathway is the primary regulator of the T3SS in *P. syringae*, with HrpRS activating *hrpL* expression. HrpL then binds to the *hrp* box to induce the translation of downstream T3 effectors (*Xiao et al., 1994*; *Hutcheson et al., 2001*; *Hendrickson et al., 2000*). HrpRSL also impacts other virulence-related mechanisms, including motility, biofilm formation, siderophore production, and oxidative stress resistance, which are under the control of complicated regulatory networks, allowing *P. syringae* to infect plants (*Haefele and Lindow, 1987*; *Dasgupta et al., 2003*; *Deng et al., 2014*; *Fan et al., 2020*; *Xie et al., 2021*; *Shao et al., 2021*; *Huang et al., 2022*). However, the methylome and function of DNA methylation in *P. syringae* pathogenesis and metabolism remain largely unknown.

In bacteria, DNA methylation is the primary level of epigenetic regulation because these prokaryotes lack the histones and nucleosomes of eukaryotic cells. Bacterial DNA has three primary forms of methylation: $N^6$-methyladenine (6mA), $N^4$-methylcytosine (4mC), and $N^5$-methylcytosine (5mC). The latter is the most common type in eukaryotes, while 6mA is the dominant form in prokaryotes. DNA methylation in bacteria is mainly the product of methyltransferase (MTase) enzymes, which transfer a methyl group from *S*-adenosyl-L-methionine to various positions on target bases, depending on the modification (*Gold et al., 1963*). MTases originate from the restriction–modification (R–M) system, which protects bacterial cells from invading DNA by recognizing and cleaving specific unmethylated motifs (*Casadesús and Low, 2006*; *Wion and Casadesús, 2006*). There are three main types of R–M systems, Types I, II, and III, categorized according to the related subunits and the precise site of DNA restriction (*Bickle and Krüger, 1993*; *Loenen et al., 2014a*). Additionally, orphan MTases are an emerging group of MTases without cognate restriction that are involved in the regulation of DNA replication and gene expression (*Casadesús and Low, 2006*). The Type I R–M system contains three host-specificity determinant (*hsd*) subunits, restriction (R), modification (M), and specificity (S), which are encoded by *hsdR*, *hsdM*, and *hsdS*, respectively. Genome sequencing and bioinformatics analyses have revealed that the HsdMSR system exists in around half of all bacteria and archaea species (*Loenen et al., 2014a*). Nonetheless, the biological roles and specific motifs alongside their targets of most MTases remain unknown, especially those related to 6mA (*Blow et al., 2016*).

A new sequencing technology, single-molecule real-time (SMRT) sequencing, can detect all three forms of DNA methylation, but particularly 6mA (*Beaulaurier et al., 2019*). SMRT-seq has been applied for complete genome sequencing and insertion sequence profiling in different *P. syringae* pv. *actinidiae* (*Psa*) strains (*Ho et al., 2019*; *Poulter et al., 2018*; *Poulter et al., 2017*). In *P. aeruginosa*, Type I R–M systems, along with their specific motifs, have been identified in the model strain PAO1 and two clinical strains via SMRT-seq. These MTases are considered to play important roles in *P. aeruginosa* virulence or drug resistance (*Doberenz et al., 2017*; *Han et al., 2022*; *Li et al., 2023*).

To study the targets and functions of DNA modifications in *P. syringae*, SMRT-seq was used to identify global methylation sites and reveal their conserved and divergent functions in the three model strains in this study. We found that HsdMSR in *Psph* was involved in modulating three pathways, namely T3SS, biofilm production, and the metabolism-related gene expression of ribosomal proteins. Moreover, we found that HsdMSR-mediated transcriptional regulation depends on the full methylation of both DNA strands. Taken together, our results provide insights into the involvement of DNA methylation in the phenotypic traits of *P. syringae* and transcription regulatory mechanisms that may apply to other pathogenic bacteria.

## Results

### Genome-wide methylome profiling of three model *P. syringae* strains

According to REBASE (*Roberts et al., 2023*) database prediction, 0–2 Type I R–M systems and 3–4 Type II R–M systems exist in the three pathovars of *P. syringae*, indicating that Type I is more conserved

than Type II in this pathogen (*Supplementary file 1*). To characterize their methylomes, we performed SMRT-seq to obtain the methylation patterns on a genome-wide scale using DNA extracted from the stationary phase of the three wild-type (WT) strains. Our results revealed a total of 10,302 modified bases, of which 3849, 4646, and 1807 nucleotides were significantly identified as being 6mA, 5mC, and 4mC modified, with an interpulse duration ratio (IPD) of >1.5 throughout the genome of *Psph* (*Figure 1A*). After identifying the significantly modified sites, we orientated the gene locus tags within the methylated bases. Most genes harbored only 6mA modifications, although 339 genes harbored all three DNA methylation types (*Figure 1B*).

In *Pst*, a total of 6002, 2242, and 3514 methylation sites were significantly identified as 6mA, 4mC, and 5mC, respectively, using the same cut-off (IPD >1.5) (*Figure 1C*). Among these modified bases, more genes equipped with all three modification types were detected in *Pst* than *Psph*, with 591 (17.7%) genes exhibiting modifications in *Pst* sequences (*Figure 1D*). Additionally, the methylome atlas of *Pss* revealed a lower incidence of methylation than those of *Psph* and *Pst*, particularly in terms of 6mA modifications, which were only seen in 457 significant 6mA occurrences under the same threshold (IPD >1.5) and a total of 2853 and 1438 methylation sites were detected as 5mC and 4mC, respectively (*Figure 1E*). Comparative analysis of the three methylations in *Pss* showed that the highest numbers of genes were modified by 4mC and 5mC simultaneously (*Figure 1F*).

## Characteristics of modified loci and GC contents in *P. syringae*

To further uncover the distribution of the modifications throughout the genome, we calculated how many of the three kinds of modified sites were in these three strains. The majority of modifications occurred in coding sequence (CDS) regions, accounting for at least 80% (*Figure 1—figure supplement 1A*), and less than 20% were in intergenic regions and non-coding RNA (tRNA and rRNA). However, compared with 4mC and 5mC, 6mA was more likely to be in intergenic regions, which suggests that it potentially functions in transcription regulation in *P. syringae*.

It is known that 5mC occurs within CpG sites in CpG islands, which contain higher GC percentages that is typical in the human genome (*Bird, 2002*). Additionally, the GC architecture can influence DNA methylation in eukaryotic cells (*Gelfman et al., 2013*). To determine the GC content characteristics of modification sites in *P. syringae*, we extracted the sequences 50 bp upstream and downstream from the modified bases and calculated their GC content. The distribution of modification sites and their GC content are shown in density plots. Compared with 4mC and 5mC, the GC contents of 6mA sites were the lowest and the closest to the average GC percentage of *P. syringae* (58% for *Psph* and *Pst*, 59.2% for *Pss*) (*Figure 1—figure supplement 1B*). In contrast, 5mC modification bases had the highest GC content, especially in *Pss* (*Figure 1—figure supplement 1C*). Furthermore, 4mC sites in *Psph* had a higher GC content than those in *Psa* and *Pss* (*Figure 1—figure supplement 1D*). Similar phenomena have been observed in various other bacterial species. For instance, the *Escherichia coli* MTase Dcm can catalyse the 5′CCWGG3′ motif (*Militello et al., 2012*). Furthermore, in *Spiroplasma* sp. strain MQ-1, more than 95% of 5mC modifications are found in high-CG sequences, which is similar to the pattern in eukaryotes (*Nur et al., 1985*).

## Methylated genes are functionally conserved among three *P. syringae* strains

When we compared the three methylation types in the three tested strains, we observed the conservation pattern among them (*Figure 2A, B* and *Supplementary file 2*). Notably, about 25–45% of methylated genes were conserved in two and three strains. Interestingly, 5mC had the highest conservation level (39.1%), which might be explained by the rare occurrence of 6mA in *Pss*. Conversely, an obviously higher degree of conservation was observed in virulence-related genes, including T3SS and alginate biosynthesis-related genes, between *Psph* and *Pst* (*Supplementary file 2*). Additionally, some methylated genes (*n* = 739) harbored sites for different modification types in the three strains (*Figure 2A*). For example, the modification sites of a Cro/CI family transcriptional factor (TF) in PSPPH_1319 were for different modification types in the three *P. syringae* strains. The Cro/CI are important TFs present in phages. The interaction between Cro and CI affects bacteria immunity status in Enterohemorrhagic *Escherichia coli* (EHEC) strains (*Jin et al., 2022*). *Psph* carried 6mA and 5mC, while *Pst* and *Pss* had 5mC and 4mC. This difference might result from differences in the MTases and their specific motifs (*Supplementary file 2*).

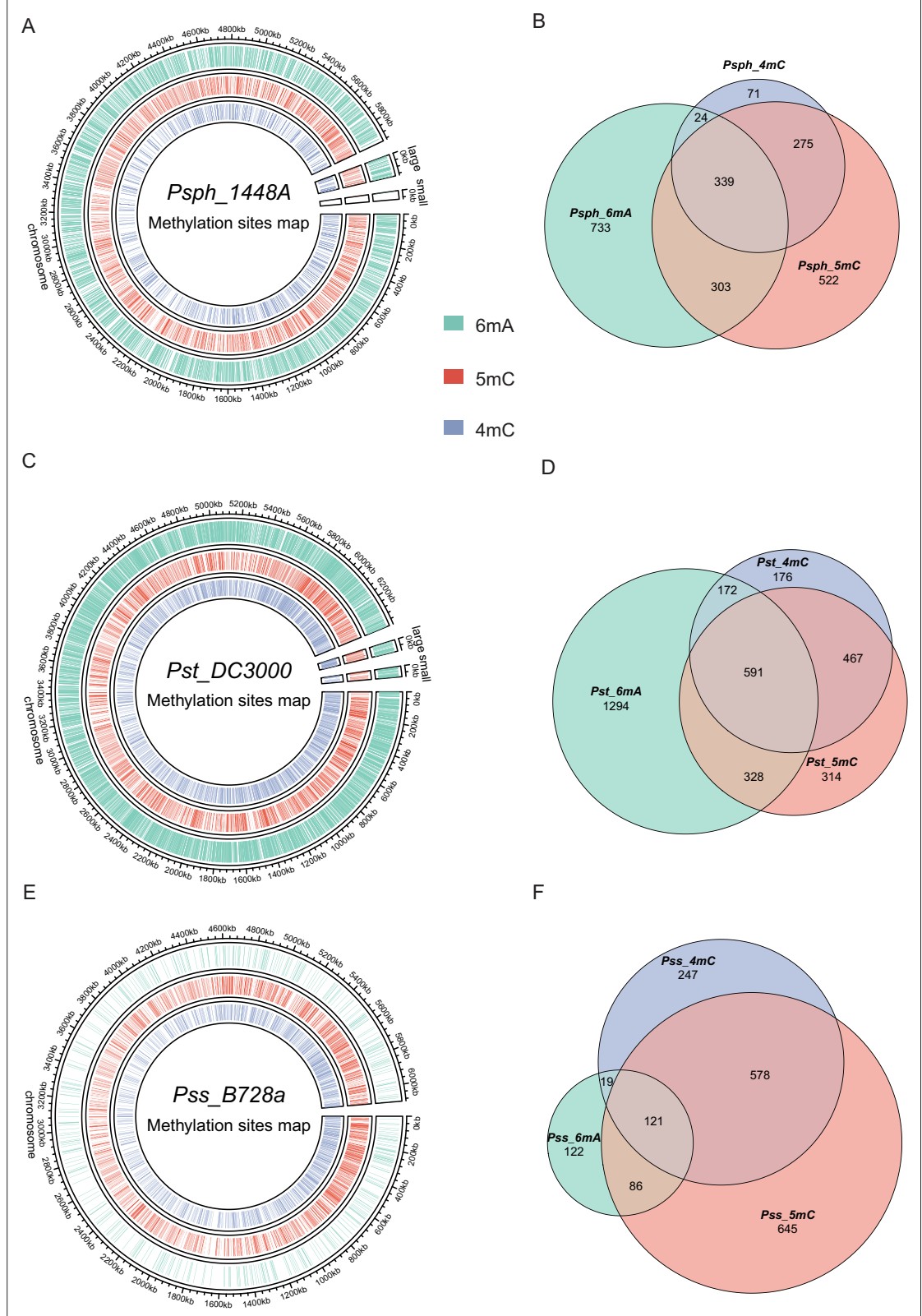

**Figure 1.** Genome-wide identification of *P. syringae* DNA methylation. (**A**) The circle map displays the distribution of 6mA, 5mC, and 4mC in *Psph* WT. (**B**) The Venn plot reveals overlapped genes within three types of DNA methylation of *Psph* WT. (**C**) The circle map displays the distribution of 6mA, 5mC, and 4mC in *Pst* WT. (**D**) The Venn plot reveals overlapped genes within three types of DNA methylation of *Pst* WT. (**E**) The circle map displays the distribution of 6mA, 5mC, and 4mC in *Pss* WT. (**F**) The Venn plot reveals overlapped genes within three types of DNA methylation of *Pss* WT.

*Figure 1 continued on next page*

Figure 1 continued

The online version of this article includes the following figure supplement(s) for figure 1:

**Figure supplement 1.** Distribution patterns of methylation sites in three model strains.

To elucidate the functions of genes methylated with these three modification types, functional enrichment analyses were performed based on gene ontology (GO) and Kyoto Encyclopedia of Genes and Genomes (KEGG) databases (*Ashburner et al., 2000*; *Kanehisa and Goto, 2000*). Analysis revealed the shared functional characteristics of genes with 5mC and 4mC modifications in *Pss* and *Pst*. In contrast, genes with 6mA modifications exhibited more conserved functional terms between *Psph* and *Pst* (*Figure 2C*). For instance, *Pst* and *Pss* contained 4mC-methylated genes enriched in signalling transduction and TF binding, whereas *Psph* exhibited enrichment in oxidation–reduction process and nucleic acid binding genes. Remarkably, notable conservation of functional terms was observed for genes with 6mA modifications, with at least 21 terms conserved between *Psph* and *Pst*. These terms included ATP binding, catalytic activity, integral component of membrane, transferase activity, and phosphorylation. We also conducted Cluster of Orthologous Group (COG) protein function analysis (*Cantalapiedra et al., 2021*; *Huerta-Cepas et al., 2019*), which assigned the methylated genes from the three strains to 20 categories with diverse functions (*Figure 2—figure supplement 1A–C*). In *Psph*, the top three COG classifications encompassed inorganic ion transport, amino acid transport, and transcription (*Figure 2—figure supplement 1A*). Furthermore, the abundance of modified genes related to cell wall/membrane/envelope biogenesis (M category) in *Pss* was lower than that of *Psph* and *Pst*. Despite the slight variations in COG distribution, the overall pattern exhibited considerable similarity and conservation across the three strains.

## Six conserved sequence motifs of 6mA and 4mC were identified in *P. syringae*

Obtaining the whole methylomes of the three strains led us to investigate the associated MTases and their target motifs. The PacBio motif finder and MEME suite (*Bailey et al., 2015*) were used to determine the specific sequence motifs within 50 bp upstream and downstream of the significantly modified bases. Using the results of both software packages with corresponding cut-offs, we identified two 6mA motifs in *Psph*: C**A**GCN$_{(6)}$CTC and RAGT**A**CTY (*Figure 3A*). The bolded bases indicate the methylation sites in these motifs. The two motifs occurred a total of 2998 and 300 times in double strands, and more than 98% and 77% of those occurrences were methylated, respectively, revealing the high methylation rates of these two motifs and implying their crucial roles in *Psph* (*Figure 3B*). However, the analysis did not obtain any identified motifs for cytosine modifications.

Three motifs for 6mA and one motif for 4mC were identified in *Pst* (*Figure 3C*). The first 6mA motif, CAARG**A**A, was found 2022 times in the genome, with 2019 of the occurrences methylated (99.8%) (*Figure 3D*). The second 6mA motif, G**A**AN$_{(4)}$RTRCC, was fully methylated in both strands. Similarly, the last 6mA motif, GY**A**GN$_{(5)}$CTRC, exhibited a high methylation level (96%) in the genome of *Pst*. Conversely, the only motif identified for 4mC in *Pst* displayed a considerably lower methylation level than the 6mA sites, with 2207 occurrences of methylation out of a total of 8048 sites (approximately 27%). Notably, no credible motifs were found in *Pss* because of its noticeably lower modification levels compared with the other two strains. Overall, the application of SMRT-seq to the three model *P. syringae* strains identified specific sequence motifs, including 6mA motifs, exhibiting extensive methylation statuses.

## *In vivo* validation revealed the activity and specificity of Type I R–M system MTase in *Psph*

The first motif, C**A**GCN$_{(6)}$CTC, identified in *Psph* is similar to the counterpart modified by a Type I R–M MTase in *P. aeruginosa* (*Beaulaurier et al., 2019*; *Figure 3—figure supplement 1*), suggesting that the putative HsdMSR is responsible for the observed motif. As no other motifs exhibited a discernible association with the Type II R–M MTases, we opted to focus our investigation on the putative HsdMSR in *Psph*. To determine the potential functions of the putative MTases, we constructed MTase knockout mutants and complementary strains of HsdMSR. Growth curve experiments showed the adverse effects on Δ*hsdMSR* compared to the WT strain (*Figure 3—figure supplement 2A*). To confirm the

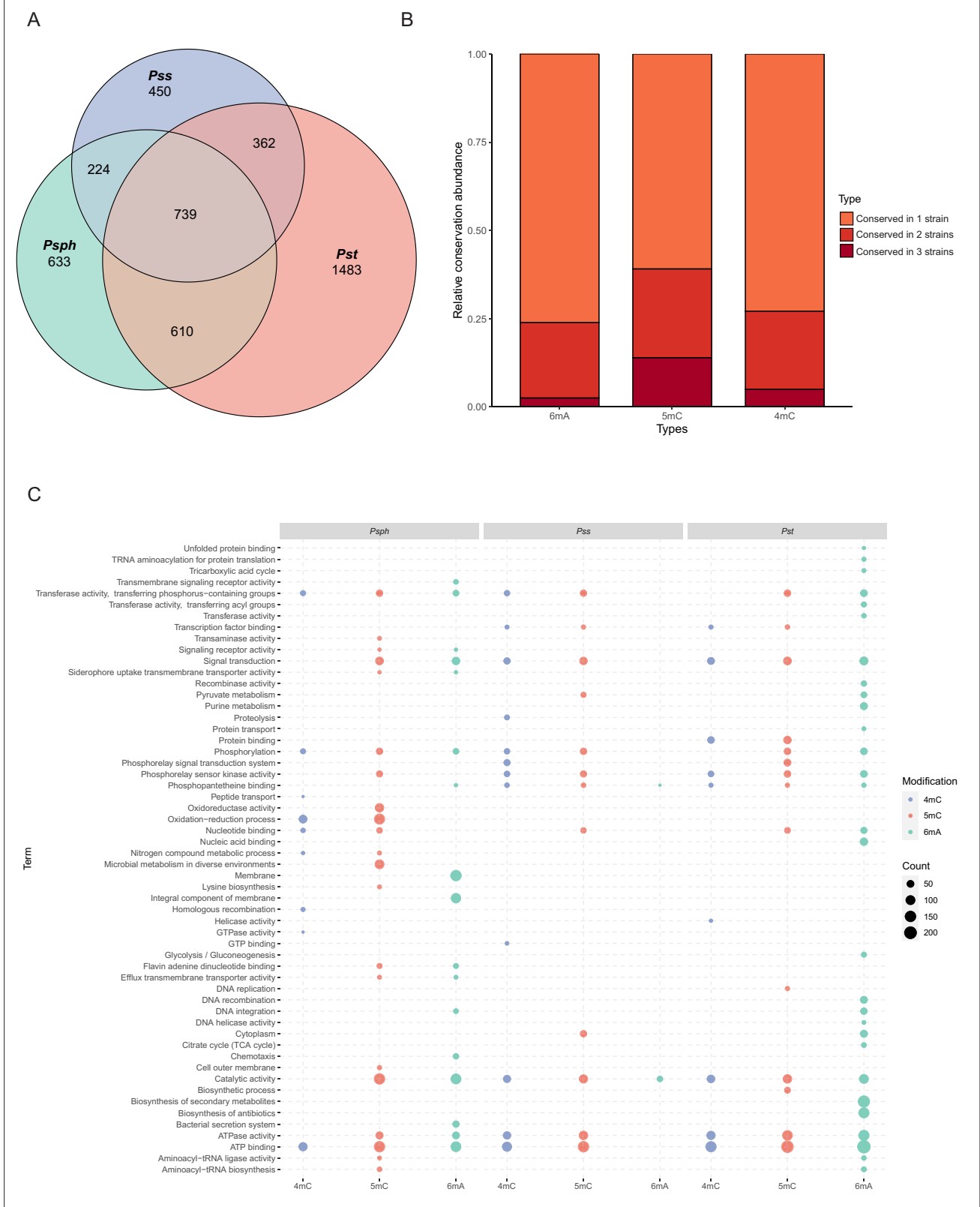

**Figure 2.** Functional enrichment analysis of methylation sites in three *P. syringae* strains. (**A**) Repartition of the total pool of modified genes among strains. (**B**) Proportion of methylated genes detected in one, two, or three genomes for all *P. syringae* strains and conserved DNA methylation sites with detected genes. (**C**) The dot plot revealed the significantly enriched functional pathways in gene ontology (GO) and Kyoto Encyclopedia of Genes and Genomes (KEGG) databases among three *P. syringae* strains. The specific names of each pathway were listed on the left, and each column with dots

*Figure 2 continued on next page*

*Figure 2 continued*

indicated the number of genes within one kind of methylation in one of three *P. syringae* strains. The size of the dots indicates the number of related genes.

The online version of this article includes the following figure supplement(s) for figure 2:

**Figure supplement 1.** Cluster of Orthologous Group (COG) analysis of methylation sites in *P. syringae*.

activity of HsdMSR, a dot blot assay was used to detect the 6mA intensity in WT and Δ*hsdMSR*. The results shown in *Figure 3—figure supplement 2B* illustrated that 6mA levels were significantly decreased in Δ*hsdMSR* compared to the WT, but were restored in the complemented strain. In addition, SMRT-seq was performed on Δ*hsdMSR*, which further profiled methylation patterns in Δ*hsdMSR* (*Figure 3—figure supplement 2C, D*). As expected, we found almost all of the reduction 6mA sites in the Δ*hsdMSR* were from motif CAGCN$_{(6)}$CTC (*Figure 3—figure supplement 2E, F*). We also noticed that 5mC and 4mC sites in the mutant were relatively similar to that in WT (*Figure 3—figure supplement 2E*), and the slight difference might be caused by sequencing errors. Overall, we propose that HsdMSR only catalyze the 6mA located on the motif CAGCN$_{(6)}$CTC, but does not affect other 6mA sites and other modification types.

In accordance with previous studies showing that growth conditions affect the bacterial methylation status (*Gonzalez and Collier, 2013*; *Krebes et al., 2014*; *Sánchez-Romero and Casadesús, 2020*), we applied dot blot experiments using the same amount of DNA (1 μg) from these three *P. syringae* strains to detect the 6mA levels during both logarithmic and stationary phases. The results revealed that 6mA levels in the stationary phase were much higher than those in the logarithmic phase in *Psph* and *Pst*, but no significant change in *Pss* (*Figure 3—figure supplement 3A*). Additionally, we found that during the stationary phase, 6mA methylation levels in *Psph* and *Pst* were higher than those in *Pss*. These findings were consistent with the MTases predication on these three strains since *Pss* does not harbor any Type I R–M systems, which are important for 6mA medication in bacteria. We also overexpressed HsdM in *Pst* and performed additional experiments in WT and the HsdM overexpression strains, including dot blot and growth curve assays. The results showed that the MTase overexpressed strain presents a higher 6mA level than WT during the logarithmic phase, and the overexpression of MTase had no effects on growth in *Pst* (*Figure 3—figure supplement 3B, C*). Taken together, the results demonstrated that 6mA levels change with the bacterial growth phase in *Psph* and *Pst*, and HsdMSR is responsible for maintaining 6mA sites within the sequence motif of C**A**GCN$_{(6)}$CTC in *Psph*.

## Transcriptomic analysis profiling of differentially regulated genes associated with virulence and metabolism in the HsdMSR mutant

To explore the regulatory influence of HsdMSR on gene expression, RNA-seq was applied to analyze WT and Δ*hsdMSR* in the stationary growth phase, which ensured that the strains had sufficient DNA methylation levels. Differentially expressed genes (DEGs) were obtained with the cut-off of |log$_2$FC| > 1 and an adjusted p-value <0.05. We identified 395 DEGs between WT and Δ*hsdMSR* under these experimental conditions. Among these, 218 genes were upregulated, while 177 genes were downregulated compared with the WT (*Figure 4A* and *Supplementary file 3*). To investigate the functional differences between WT and Δ*hsdMSR*, GO and KEGG databases were used to perform functional enrichment analyses based on the DEGs. When using adjusted p-value <0.05 as a significant cut-off, the upregulated functional terms included alginate biosynthesis, fructose and mannose metabolism, and the oxidation–reduction process (*Figure 4B*). The downregulated pathways included the citrate cycle, oxidative phosphorylation, ribosome structure, and translation (*Figure 4B*). A total of 116 genes showed bigger differences (|log$_2$FC| > 2) except for genes related to ribosomal protein, T3SS, and alginate synthesis. Among these genes, 31 were annotated as hypothetical proteins and 4 as transcription factors with unknown functions, and the remaining genes mostly encoded metabolism-related enzymes. These enzymes might have effects on growth defects in Δ*hsdMSR*.

We also analyzed the function alteration via COG category analysis of DEGs, which revealed that the upregulated genes were more likely to be involved in lanes E (amino acid transport and metabolism), G (carbohydrate transport and metabolism), and I (lipid transport and metabolism) (*Figure 4C*). Downregulated DEGs were significantly involved in lanes F (nucleotide transport and metabolism) and J (translation, ribosomal structure, and biogenesis) (*Figure 4D*). To investigate the correlation

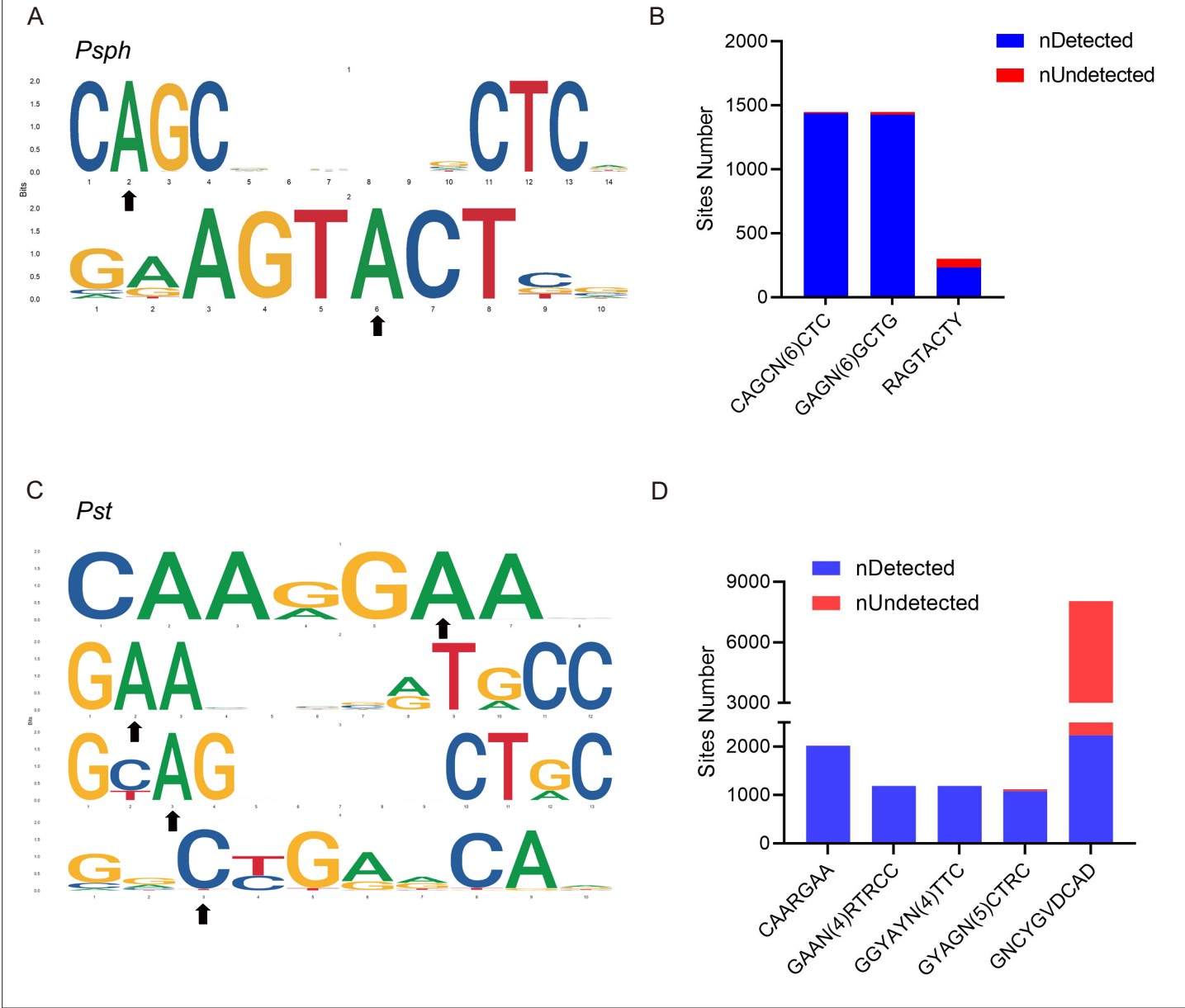

**Figure 3.** DNA methylation motifs in *P. syringae*. (**A**) 6mA methylation motifs found in *Psph* using SMRT-seq. The black arrows indicate the site of adenine methylation. (**B**) Bar plot shows the abundance of methylated numbers throughout all motif sites in *Psph*. (**C**) 6mA and 4mC methylation motifs found in *Pst* using SMRT-seq. The black arrows indicate the site of base methylation. (**D**) Bar plot shows the abundance of methylated numbers throughout all motif sites in *Pst*.

The online version of this article includes the following figure supplement(s) for figure 3:

**Figure supplement 1.** Comparative genomic analysis of Type I RM system among bacterial genus.

**Figure supplement 2.** Identification of Type I DNA methyltransferase in *Psph*.

**Figure supplement 3.** Effects of growth phases on methylation in *P. syringae*.

between DEGs and HsdM-recognizing motif sites, we performed comparative analysis. Although the overlap was not statistically significant (p > 0.05), we identified 76 overlapping genes (*Figure 4E*). Eight DEGs harbored the HsdMSR methylation motif at the putative promoter region (within 100 bp upstream of the gene) (*Supplementary file 4*).

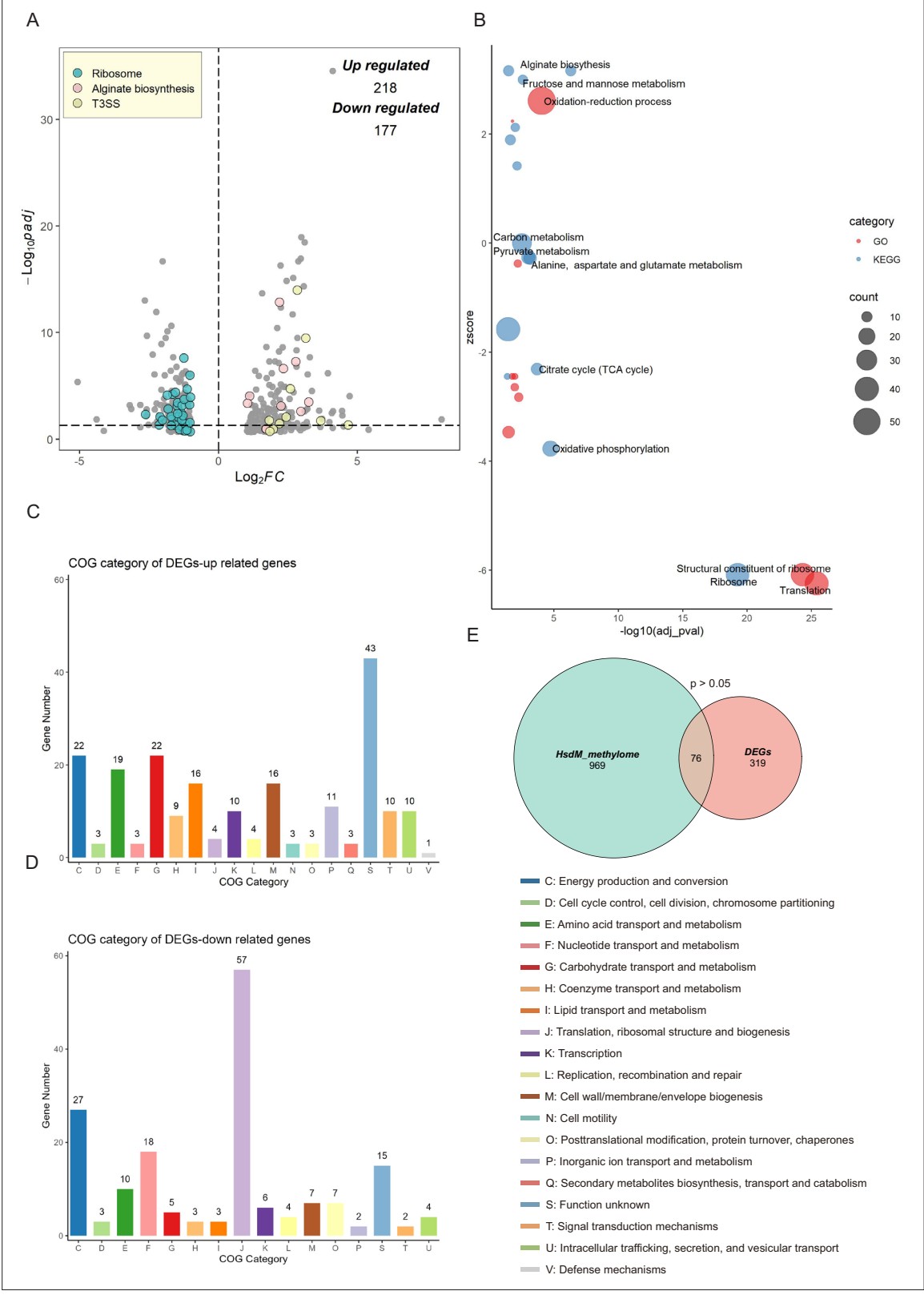

**Figure 4.** Transcriptional changes profiling of *hsdMSR* mutant in *Psph*. (**A**) The volcano plot reveals differentially expressed genes (DEGs) between *Psph* WT and Δ*hsdMSR*. The DEGs were (|log$_2$FC| > 1 and adjusted p-value <0.05) in blue (ribosomal protein), pink (alginate biosynthesis), and yellow (T3SS). Each dot represents one gene. (**B**) Bubble plot shows enriched gene ontology (GO) (red) and Kyoto Encyclopedia of Genes and Genomes (KEGG) (blue) terms based on the DEGs between *Psph* WT and Δ*hsdMSR*. The *x*-axis shows the significance of functional annotation terms (−log$_{10}$ adjusted

*Figure 4 continued on next page*

Figure 4 continued

p-value), and the *y*-axis indicates the *Z*-score of terms. The bubble size represents the gene number of terms. (**C**) Cluster of Orthologous Group (COG) classification and distribution of upregulated DEGs. (**D**) COG classification and distribution of downregulated DEGs. COG terms are highlighted in different colors. (**E**) Venn plot reveals the overlapped genes between DEGs and genes within the HsdMSR motif.

## HsdMSR was required for T3SS and biofilm formation in *P. syringae*

To confirm the RNA-seq results, we performed quantitative real-time PCR (RT-qPCR) experiments on T3SS genes. Several T3SS effector genes were identified as upregulated, including *hopAE1*, *hrpF*, *hrpA2*, *hrpK1*, and *hrpW1*. RT-qPCR was applied under the same conditions as RNA-seq, and the transcriptional levels of all genes were confirmed as being significantly higher in the Δ*hsdMSR* than the WT (*Figure 5A*). These gene expression levels were restored in the complemented strain. To determine whether the loss of HsdMSR affected the *Psph* virulence phenotypes because of alterations to T3 effectors, we allowed the WT, Δ*hsdMSR*, and complemented strains to infiltrate the primary leaves of bean plants. At 6 days post-inoculation, Δ*hsdMSR* was observed to induce more severe symptoms than the WT and complemented strains (*Figure 5B*). Overall, HsdMSR inhibited the expression of T3 effectors under KB conditions, and the loss of this modification system enhanced the pathogenicity of the strain during plant infection.

Besides T3SS, alginate biosynthesis-related genes were also observed among the DEGs between WT and Δ*hsdMSR*. Alginate biosynthesis proteins have been demonstrated to be essential for extracellular polymeric substances production, which enhances biofilm formation during the bacterial infection process (*Boyd and Chakrabarty, 1995*; *Ichinose et al., 2013*). RT-qPCR experiments were performed on the relevant genes, including *alg8*, *algE*, *alg44*, *algF*, and *algD*. The expression levels of these genes in Δ*hsdMSR* were significantly increased compared with the WT and complemented strain (*Figure 5C*). To further investigate the influence of HsdMSR on biofilm formation, we performed a crystal violet staining assay to detect biofilm production by the three strains. The intensity of crystal violet staining of biofilms formed by the WT and complemented strains was significantly less strong than that of Δ*hsdMSR* cells (*Figure 5D*), supporting the role of HsdMSR in controlling biofilm formation. We therefore concluded that HsdMSR regulates the virulence of *P. syringae* by tuning its expression of T3SS and alginate biosynthesis genes.

## HsdMSR regulated ribosomal protein synthesis and translational efficiency

We noticed that HsdMSR was important for bacterial growth (*Figure 3—figure supplement 2*), while the expression of many ribosomal protein genes was reduced with the deletion of *hsdMSR* (*Figure 5E*). Regulation of ribosomal gene expression is essential to the integrity of the ribosome structure, which affects translation (*Zhou et al., 2015*). Therefore, we hypothesized that the low expression level of ribosome proteins in Δ*hsdMSR* would result in delayed growth. We selected four genes encoding different ribosome subunits (*rplS*, *rpmD*, *rpsS*, and *rpsJ*) to verify the RNA-seq results through RT-qPCR. While *rplS* and *rpmD* encode 50S ribosomal proteins (L19 and L30), *rpsS* and *rpsJ* are responsible for 30S ribosomal proteins (S19 and S10). The RT-qPCR results demonstrated that the mRNA expression levels of these genes were significantly lower in Δ*hsdMSR* than in WT, while levels in the complemented strain were restored to the WT level (*Figure 5E*).

Given the important role of ribosomal proteins in translation, we performed ribosome profiling, also termed Ribo-seq, of the *Psph* WT and Δ*hsdMSR* strains to detect the influence of HsdMSR on translational efficiency (TE) (*Supplementary file 5*). When we compared the WT and Δ*hsdMSR*, the TE of 162 genes was significantly different ($|\log_2 FC| > 1.5$) (*Figure 5F*). Of these genes, 73 genes had enhanced TE, while 89 had suppressed TE (*Figure 5F*). Most of these genes were related to transmembrane transport and transcription regulation (*Figure 5—figure supplement 1A, B*). Remarkably, the TE of genes linked to the oxidation–reduction process tended to be suppressed (*Figure 5—figure supplement 1B*). These results showed that HsdMSR plays an important role in the regulation of metabolism by promoting ribosomal protein synthesis and modulating TE in *Psph*.

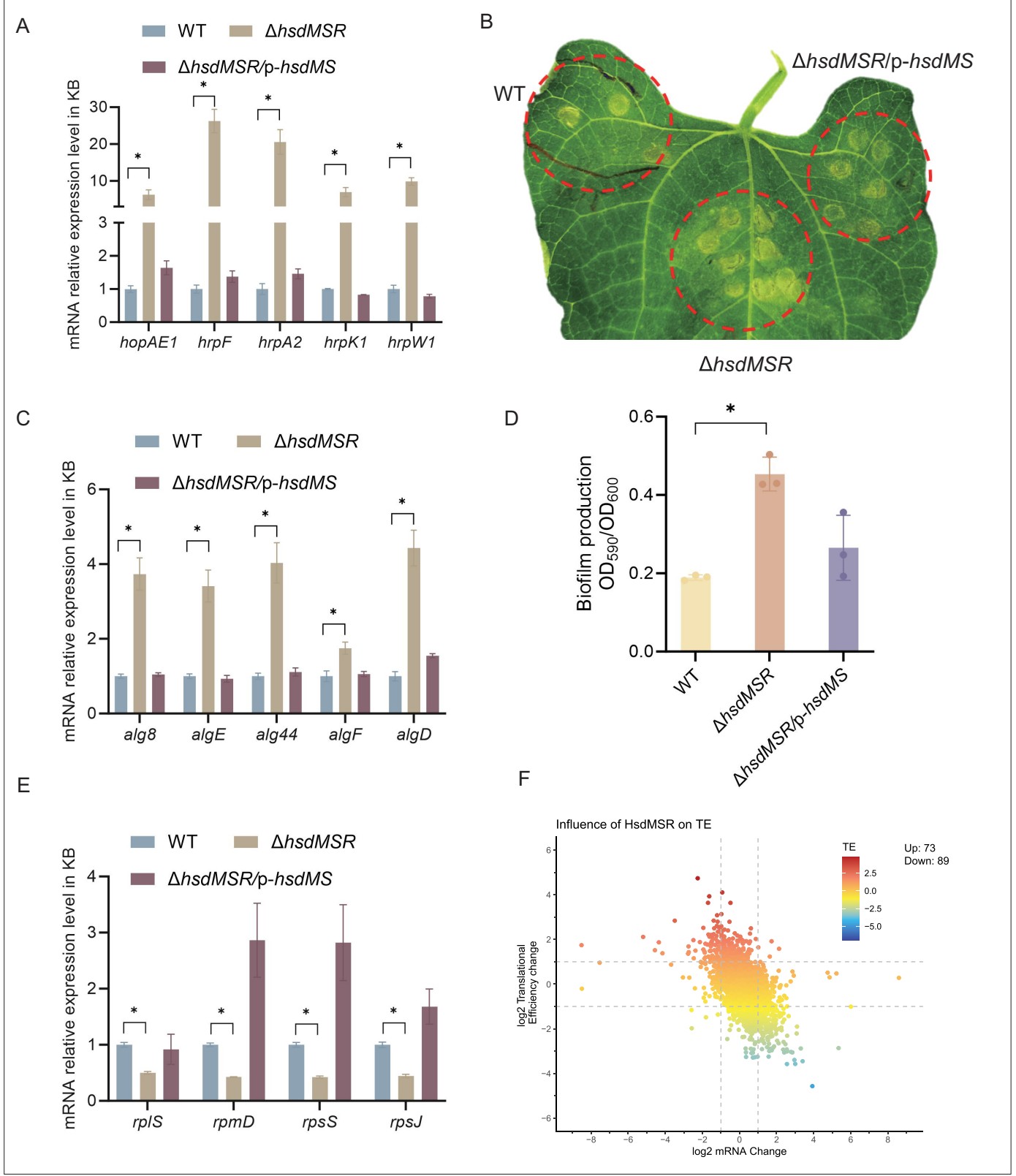

**Figure 5.** Influence of HsdMSR on virulence and metabolism in *Psph*. (**A**) HsdMSR negatively regulated T3SS-related genes. Data are shown as means ± SD (*n* = 3). Statistical significance was determined using two-tailed Student's t-test (*p < 0.05). (**B**) Disease symptoms caused by *Psph* WT, Δ*hsdMSR*, and the complemented strains, photographing 6 days after inoculation of 10⁵ CFU/ml bacteria. (**C**) HsdMSR negatively regulated alginate biosynthesis-related genes. Data are shown as means ± SD (*n* = 3). Statistical significance was determined using two-tailed Student's t-test (*p < 0.05). (**D**) The

*Figure 5 continued on next page*

*Figure 5 continued*

quantification of biofilm production in the *Psph* WT, Δ*hsdMSR*, and the complemented strains using a crystal violate staining assay. Data are shown as means ± SD (*n* = 3). Statistical significance was determined using two-tailed Student's t-test (*p < 0.05). (**E**) HsdMSR positively regulated ribosomal protein-related genes. Data are shown as means ± SD (*n* = 3). Statistical significance was determined using two-tailed Student's t-test (*p < 0.05). (**F**) The scatterplot shows the translational efficiency (TE) and mRNA change between *Psph* WT and Δ*hsdMSR*. The x-axis presents the log$_2$FC of the mRNA level, and the y-axis shows the log$_2$FC of the TE. The greater TE is represented in red, whereas the lesser TE is displayed in blue.

The online version of this article includes the following figure supplement(s) for figure 5:

**Figure supplement 1.** Number of genes with significant translational efficiency (TE) changes.

## HsdMSR regulation of *hrpF* was dependent on the full methylation of both strands

To investigate whether and how HsdMSR directly affects gene expression, we focused on those DEGs whose upstream regions harbor its motif. Interestingly, we noticed a methylation motif in the upstream region of *hrpF* (encoding the pathogenicity factor HrpF), which increased its expression level in the *Psph* WT strain (**Figures 5A and 6A**). To further explore the effects of HsdMSR on the transcription of *hrpF*, we constructed a *lux*-reporter plasmid carrying the motif and extended it by 50 bp, which covered the upstream region of *hrpF*. After transferring the plasmid into the WT and Δ*hsdMSR* strains, we detected a significant difference in the expression level of *hrpF-lux* between these two strains. The *hrpF-lux* signal in Δ*hsdMSR* increased along with culture time and reached a peak at 24 hr when the methylation level was also elevated in the late growth stage (**Figure 6B**). This result of the *lux*-reporter assay confirmed the RNA-seq and RT-qPCR results showing that *hrpF* was remarkably upregulated in Δ*hsdMSR* compared with the WT strain.

To detect the influence of the 6mA site within the HsdMSR motif on the expression of *hrpF*, we induced a point mutation on the single strand in the reporter plasmid to change the 6mA to C (C**C**GCN$_6$CTC/ CCGCN$_6$C**G**C). This resulted in a low *hrpF* expression level similar to that of the WT sequence carrying the A base (**Figure 6C**), indicating that a hemi-methylated pattern is insufficient for transcriptional alteration. To verify our hypothesis, we constructed a reporter with two A mutations (C**C**GCN$_6$C**G**C), transferred it to the WT and Δ*hsdMSR* strains, and detected signal changes. As expected, the reporter carrying the HsdMSR motif without two A bases resulted in a significantly higher signal than the hemi-mutated and non-mutated reporters in the WT (**Figure 6C**). Taken together, the results showed that HsdMSR regulation of *hrpF* transcription was based on both strands being fully methylated.

## Discussion

Recent advances in sequencing technology have expanded our understanding of DNA methylation in bacteria. Numerous DNA MTases have been extensively characterized in different pathogenic bacteria, and data on their distribution patterns, recognition motifs, and biological functions have been collated (**Han et al., 2022**; **Li et al., 2023**; **Murray et al., 2012**; **Lee et al., 2015**; **Fang et al., 2017**; **Kumar et al., 2018**). However, most research has concentrated on animal pathogenic bacteria, with limited insights into phytopathogenic bacteria except for *Xanthomonas* (**Seong et al., 2016**; **Park et al., 2019**; **Park et al., 2021**). Consequently, in this study, we employed SMRT-seq to profile the methylome patterns and specific conserved sequence motifs of three *P. syringae* strains. Notably, we discovered that the virulence and metabolism of *Psph* are regulated by Type I R–M MTase HsdM. The nonfunctional nature of the subunit restriction endonuclease HsdR was acknowledged, although its potential functionality relies on the presence of MTase (**Doberenz et al., 2017**).

Comprehensive analysis of the methylome atlas revealed that 6mA is the most prevalent modification type in *Psph* and *Pst*, mirroring the trends observed in other bacteria. Conversely, *Pss* displays the lowest occurrences of 6mA throughout its genome (**Figure 1**). This discrepancy might be attributed to the absence of the Type I R–M MTase responsible for 6mA in *Pss*. Similarly, the presence of two Type I R–M MTases in *Pst* possibly contribute to it having a higher number of 6mA sites than *Psph*. Additionally, all three pathovars had a similar pattern, with 4mC being the least frequent modification type. Despite belonging to the same *P. syringae* species, the three strains tested displayed remarkably

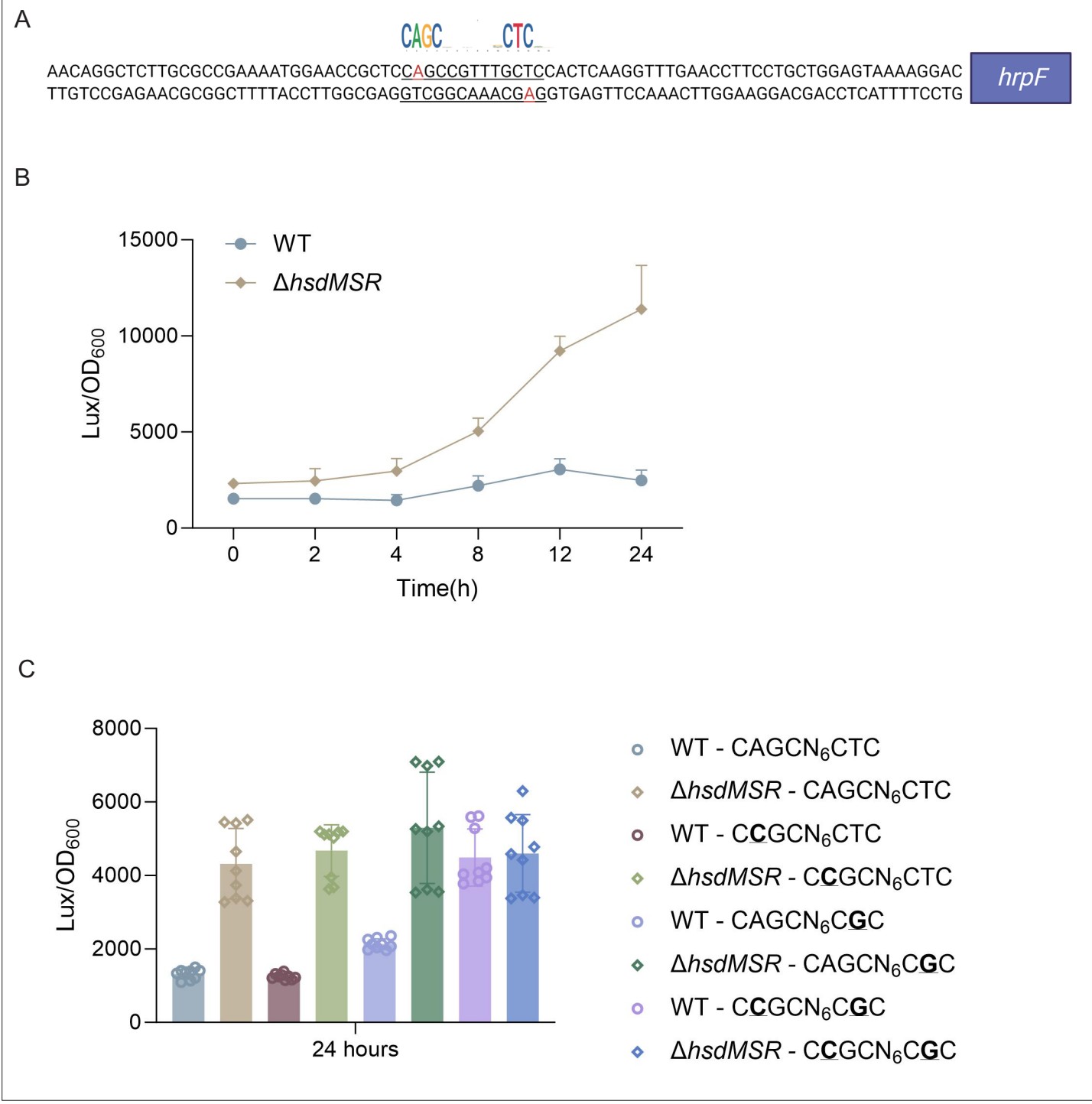

**Figure 6.** 6mA methylation regulates gene transcription based on fully methylated. (**A**) Upstream region sequence of *hrpF* carrying the HsdMSR motif. Adenine methylation is highlighted in red. (**B**) Constantly lux activity detection of the *hrpF* between *Psph* WT and Δ*hsdMSR*. Data are shown as means ± SD (*n* = 3). (**C**) Lux activity of *hrpF* between *Psph* WT and Δ*hsdMSR* with single- or double-point mutations. In the HsdMSR motif, 'A' was replaced by 'C' or 'T' was replaced by 'G', highlighted in bold and underlined. Data are shown as means ± SD (*n* = 9).

divergent methylation patterns, which is reminiscent of the phenomenon observed in *Xanthomonas* spp. (*Seong et al., 2016*).

We further identified two (both 6mA) and four (three 6mA, one 4mC) motifs in *Psph* and *Pst*, respectively, but none in *Pss* (*Figure 3*). In fact, more motifs were predicted by the MEME and PacBio

motif finders, but we chose only those motifs identified by both algorithms for higher accuracy. We found that the 6mA motifs had extremely high methylation levels, ranging from 77% to 100%, in the stationary phase of *P. syringae*. We subsequently found that the 6mA methylation levels increased during *Psph* growth, similar to the observations in *P. aeruginosa* (*Doberenz et al., 2017*). However, DNA methylation is much more stable under different conditions and growth phases in *Helicobacter pylori* and *Salmonella typhimurium* (*Krebes et al., 2014*; *Bourgeois et al., 2022*).

MTases have been reported to play important roles in bacterial growth; for example, they are involved in cell cycle processes such as chromosomal replication in *E. coli* and *Caulobacter crescentus* (*Reyes-Lamothe and Sherratt, 2019*; *Gonzalez et al., 2014*; *Kozdon et al., 2013*). DNA methylation can activate DnaA and signal the initiation of chromosome replication to affect the growth of gamma-proteobacteria (*Seong et al., 2021*; *Boye et al., 1996*). Previous studies revealed that DNA methylation influences bacterial growth through genes related to diverse carbohydrate transport mechanisms (*Park et al., 2019*). We report that the 6mA modification catalyzed by HsdM in *Psph* positively affects its growth, which can be explained by the downregulation of ribosomal proteins and the alteration of metabolism-related gene TE. However, many MTases are thought to be uncorrelated to growth in bacterial species such as *P. aeruginosa* and *Klebsiella pneumoniae* (*Doberenz et al., 2017*; *Han et al., 2022*; *Fang et al., 2017*).

Many MTases and DNA methylations are involved in bacterial virulence through their modulation of gene expression (*Doberenz et al., 2017*; *Han et al., 2022*; *Fang et al., 2017*; *Kumar et al., 2018*; *Park et al., 2019*; *Park et al., 2021*; *Manzer et al., 2023*; *Banas et al., 2011*; *Oliveira et al., 2020*). In this study, RNA-seq analysis of the *hsdMSR* mutation strain revealed 395 DEGs. Remarkably, not all DEGs harbored the HsdMSR methylation motif in their promoter region. There were no significant correlations between the HsdM-recognized motif sites in the promoter region and DEGs, even though DNA methylation is generally known to affect gene expression by altering the interactions between DNA and proteins such as TFs, which compete with MTases at specific motif sites and thus influence downstream transcription (*Lim and van Oudenaarden, 2007*). However, recent studies have revealed that DNA methylation within CDS can also alter gene expression in bacteria; for example, 5mC in CDS can enhance transcription while blocking the transcription of CpG islands in promoters in eukaryotic cells (*Kahramanoglou et al., 2012*; *Rishi et al., 2010*). We hypothesize that 6mA in *Psph* can regulate gene expression directly and indirectly. We confirmed that the 6mA motif in the promoter region of *hrpF* can directly and negatively regulate gene transcription in a manner dependent on full methylation of the strands (*Figure 6*). In addition, DNA methylation can change DNA curvature to decrease the thermodynamic stability of the double helix (*Diekmann, 1987*). Therefore, those DEGs carrying modified sites, including alginate biosynthesis-related genes and T3 effectors, are believed to be caused by the alteration of nucleoid topology, as seen in *Salmonella* and *H. pylori* (*Kumar et al., 2018*; *Camacho et al., 2005*). Furthermore, the 6mA sites, including those in virulence-related *hrc/hrp* and *alg* genes, are conserved in *Psph* and *Pst*, implying that similar phenotypes occur in the latter strain. Altogether, 6mA modifications in *Psph* were observed to act as important epigenetic regulators of gene expression.

Apart from the established roles of 6mA and HsdMSR in *P. syringae*, certain signals or factors may influence HsdMSR expression. For instance, we confirmed that the growth phase affects methylation levels in *P. syringae*. Previous studies have shown that increased temperatures can reduce methylation levels, as observed in *P. aeruginosa* PAO1 (*Doberenz et al., 2017*). These findings suggest that HsdMSR expression may be responsive to both intrinsic cellular states and extrinsic environmental conditions. To further explore potential upstream TFs regulating the expression of HsdMSR, we searched for TF-binding sites in the *hsdMSR* promoter using our own databases (*Fan et al., 2020*; *Shao et al., 2021*; *Sun et al., 2024*). As a result, we found three candidate TFs (PSPPH_0061, PSPPH_3268, and PSPPH_3504) that might directly bind and regulate HsdMSR expression. Future studies on these TFs and their interactions with the HsdMSR promoter would help clarify the regulatory network governing HsdMSR activity.

Moreover, R–M systems are known for their intrinsic role as innate immune systems in anti-phage infection, and present in around 90% of bacterial genomes (*Oliveira et al., 2014*). R–M systems protect bacteria self by recognizing and degrading foreign phage DNA via methylation-specific site and restriction endonucleases (REases) (*Loenen et al., 2014b*). In addition, other phage-resistance systems are similar to R–M systems but carry extra genes. One is called the phage growth limitation

(Pgl) system, which modifies and cleaves phage DNA. However, the Pgl only modifies the phage DNA in the first infection cycle, and cleaves phage DNA in the subsequent infections, which gives a warn to the neighboring cells (*Hampton et al., 2020*; *Hoskisson et al., 2015*). To counteract R–M and R–M--like systems, phages have evolved strategies, including unusual modifications such as hydroxymethylation, glycosylation, and glucosylation. They can also encode their own MTases to protect their DNA or employ strategies to evade restriction systems and other anti-RM defenses (*Vasu and Nagaraja, 2013*; *Murphy et al., 2013*; *Iida et al., 1987*).

As more studies apply SMRT-seq to investigate bacterial methylomes, it has become evident that the epigenetic regulation of gene expression is highly prevalent among bacteria. Understanding the mechanisms through which methylation functions can provide novel insights into how strain-specific epigenetic modifications shape the adaptive responses of bacteria to distinct environmental challenges. As the repertoire of MTases varies among *P. syringae* strains, this approach will help us to better understand the diversity of DNA methylation and epigenetic patterns among *P. syringae* species.

# Materials and methods

## Bacterial strains and culture conditions

The bacterial strains, plasmids, and primers used in this study can be found in *Supplementary file 6*. All three model strains and mutations were cultured at 28°C in KB medium with shaking at 220 rpm or Luria-Bertani (LB) agar plates. The *E. coli* strains were grown at 37°C in LB broth with shaking at 220 rpm or on LB agar plates. Antibiotics were used at the following concentrations: kanamycin at 50 µg/ml, spectinomycin at 50 µg/ml, and rifampin 25 µg/ml.

## DNA extraction and SMRT sequencing

Three WT model strains and *hsdMSR* knockout *Psph* strains were cultured to the stationary phase. The genomic DNA of the strains was extracted using a TIANamp Bacteria DNA kit (Tiangen Biotech), using the manufacturer's standard protocols. The SMRT sequence was performed at Abace Biotechnology company using Pacific Biosciences sequel II and Ile sequencer (PacBio, Menlo Park, CA, USA). SMRT-seq reads were aligned to the genome reference of *Psph* 1448A (GCF_000012205.1), *Pst* DC3000 (GCF_000007805.1), and *Pss* B728a (GCF_000012245.1), respectively. SMRTLink software v13.0 was used to perform DNA methylation analysis. A modification quality value score of 50 and 100 was used to call the modified bases A and C, respectively.

## Deletion mutant and complemented strains construction

The restriction enzymes digested the pK18mobsacB suicide plasmid (*Kvitko and Collmer, 2011*) listed in *Supplementary file 6*. The upstream (~1500bp) and downstream (~1000bp) fragments of *hsdMSR* open reading frame (ORF) were amplified from the *Psph* genome and digested. Then, the digested upstream and downstream fragments were ligated with T4 DNA ligase (NEB). The ligated fragments were inserted into the digested plasmid using ClonExpress MultiS One Step Cloning Kit (Vazyme). The recombinant plasmids were transformed into the *Psph* WT strain in the KB plate with rifampin and kanamycin. The single colonies were picked to a sucrose plate and then cultured in both KB with kanamycin/rifampin and KB with rifampin alone. Loss of kanamycin resistance indicated a double crossover. Finally, the deletion strains were further confirmed by qRT-PCR to detect the mRNA level of *hsdMSR*. For the complemented plasmids, the ORF of *hsdMS* was amplified by PCR from the *Psph* genome and cloned into the pHM1 plasmid.

## Growth curve measure assay

Overnight cultures of *Psph* and Δ*hsdMSR* were diluted to an $OD_{600}$ of 0.1 in fresh KB liquid medium for use as the inoculum. One hundred µl of the inoculum was aliquoted into a 96-well microtiter plate in triplicate and incubated at 28°C for growth. $OD_{600}$ values were recorded per 2 hr for eight times for plotting of growth curves.

## Dot blotting assay

Dot blotting was performed as previously described (*Xie et al., 2018*). Briefly, DNA samples were denatured at 95°C for 10 min and cooled down on ice for 3 min. Samples were spotted on the

nylon membrane and air dry for 5 min, followed by heat-crosslink at 80°C for 2 hr. Membranes were blocked in 5% nonfat dry milk in TBST for 1 hr at room temperature, incubated with $N^6$-mA antibodies (1:1000) overnight at 4°C. After five washes with TBST, membranes were incubated with HRP-linked secondary anti-rabbit IgG antibody (1:5000) for 1 hr at room temperature. Signals were detected with ChemiDoc Imaging Systems (Bio-Rad). After imaging, incubate the membrane with methylene blue staining buffer for 15 min with gentle shaking.

### RNA-seq analyses

*Psph* and *ΔhsdMSR* strains were first cultured to the stationary phase in KB medium at 28°C. Cultures were collected, and total RNA was extracted using the RNeasy mini kit (QIAGEN) following the manufacturer's protocol and genomic DNA contamination was removed by DNaseI treatment (NEB). Subsequently, rRNA was depleted by using the MICROBExpress kit (Ambion), and the remaining mRNA was used to generate the cDNA library according to the NEBNext UltraTM II RNA Library Prep Kit protocol (Illumina), which was then sequenced using the Illumina HiSeq 2000 system, generating 150 bp paired-end reads. Two biological replications have been performed. For each RNA-seq sample, raw sequencing reads were quality-trimmed using trim_galore (version 0.6.7) and aligned to the *Psph* genome using hisat2 (version 7.5.0)(*Kim et al., 2015*). DEGs were identified using DESeq2 (*Love et al., 2014*), and function enrichment analysis of DEGs was conducted using the R package clusterProfiler (*Yu et al., 2012*).

### Ribo-seq library construction and analysis

The construction of the Ribo-seq library followed the previous protocol (*Hua et al., 2022*). Briefly, bacteria were cultured to an $OD_{600}$ of 0.4, at which point chloramphenicol was added to a final concentration of 100 μg/ml for 2 min. Cells were then pelleted and washed with pre-chilled lysis buffer [25 mM Tris-HCl, pH 8.0; 25 mM $NH_4Cl$; 10 mM $Mg(OAc)_2$; 0.8% Triton X-100; 100 U/ml RNase-free DNase I; 0.3 U/ml Superase-In; 1.55 mM chloramphenicol; and 17 mM GMPPNP]. The pellet was resuspended in lysis buffer, followed by three freeze–thaw cycles using liquid nitrogen. Sodium deoxycholate was then added to a final concentration of 0.3% before centrifugation. The resulting supernatant was adjusted to 25 $A_{260}$ units and mixed with 2 ml of 500 mM $CaCl_2$ and 12 μl MNase, making up a total volume of 200 μl. After the digestion, the reaction was quenched with 2.5 ml of 500 mM EGTA. Monosomes were isolated using Sephacryl S400 MicroSpin columns, and RNA was purified using the miRNeasy Mini Kit (QIAGEN). rRNA was removed using the NEBNext rRNA Depletion Kit, and the final library was constructed with the NEBNext Small RNA Library Prep Kit. For each sample, ribosome footprint reads were mapped to the *Psph* 1448A reference genome, and the TE was calculated by dividing the normalized Ribo-seq counts by the normalized RNA counts. Two biological replicates were performed for all experiments.

### RT-qPCR verification

For RT-qPCR, RNA was purified using the RNeasy minikit (QIAGEN). The cDNA synthesis was performed using a FastKing RT Kit (Tiangen Biotech). The assay was performed by SuperReal Premix Plus (SYBR Green) Kit (Tiangen Biotech) according to the manufacturer's instruction. Each sample was repeated thrice with 600 ng cDNA and 16S rRNA as the internal control. The fold change represents the relative expression level of mRNA, which can be estimated by the threshold cycle (Ct) values of $2^{-(\Delta\Delta Ct)}$.

### Plant infection assay

The bean (*P. vulgaris* cv. *Red Kidney*) plants were used for the pathogenicity assay. The plant was grown in a greenhouse, as described previously (*Xiao et al., 2007*). Overnight bacterial cultures were diluted to $OD_{600} = 0.2 \times 10^{-3}$ and were hand-inoculated into the primary leaves of week-old bean plants for 6 days.

### Biofilm formation assay

Biofilm production was detected, as previously reported (*Shao et al., 2019*). In brief, overnight bacterial cultures of *Psph* WT and *hsdMSR* mutants were diluted into $OD_{600} = 0.1$ and transferred to a 10-ml borosilicate tube containing 1ml KB medium. Then, the cultures grow statically at 28°C

for 3–5 days. Then, 0.1% crystal violet was used to stain the biofilm adhered to the tube tightly for 30 min, and other components bound to the tube loosely were washed off with distilled deionized water. The remaining crystal violet was fully dissolved in 1 ml 95–100% ethanol with constant shaking. Then, it was transferred to a transparent 96-well plate to measure its optical density at 590 nm using a Biotech microplate reader. The experiment was repeated using three independent bacterial cultures.

## Data analysis and statistics

The graphs in this paper were plotted using ggplot2 in R 4.2.0 software and GraphPad Prism 10.0.3 (GraphPad Inc) Differences between groups were analyzed using Student's two-tailed $t$-tests. The results of all statistical analyses are shown as mean ± SD. All experiments were repeated independently at least three times with similar results.

## Acknowledgements

This study was supported by grants from Guangdong Major Project of Basic and Applied Basic Research (2020B0301030005), Shenzhen Science and Technology Fund (JCYJ20210324134000002), the National Natural Science Foundation of China (32172358), General Research Funds of Hong Kong (11103221, 11101722, and 11102223), and the Sichuan Province Science and Technology Program (2024ZYD0134). The funders had no role in study design, data collection, interpretation, or the decision to submit the work for publication.

## Additional information

### Funding

| Funder | Grant reference number | Author |
|---|---|---|
| Guangdong Major Project of Basic and Applied Basic Research | 2020B0301030005 | Xin Deng |
| Shenzhen Science and Technology Innovation Program | JCYJ20210324134000002 | Xin Deng |
| National Natural Science Foundation of China | 32172358 | Xin Deng |
| Sichuan Science and Technology Program | 2024ZYD0134 | Xin Deng |
| General Research Funds of Hong Kong | 11103221 | Xin Deng |
| General Research Funds of Hong Kong | 11102223 | Xin Deng |
| General Research Funds of Hong Kong | 11101722 | Xin Deng |

The funders had no role in study design, data collection and interpretation, or the decision to submit the work for publication.

### Author contributions

Jiadai Huang, Conceptualization, Resources, Data curation, Formal analysis, Supervision, Validation, Investigation, Visualization, Methodology, Writing – original draft, Project administration, Writing – review and editing; Fang Chen, Data curation, Formal analysis, Validation, Investigation, Methodology, Writing – review and editing; Beifang Lu, Formal analysis, Writing – review and editing; Yue Sun, Validation, Methodology, Writing – review and editing; Youyue Li, Validation, Writing – review and editing; Canfeng Hua, Conceptualization, Methodology; Xin Deng, Conceptualization, Supervision, Funding acquisition, Methodology, Project administration, Writing – review and editing

## Author ORCIDs

Jiadai Huang (iD) https://orcid.org/0000-0002-4104-6836
Yue Sun (iD) http://orcid.org/0009-0008-2843-0017
Youyue Li (iD) https://orcid.org/0000-0003-4976-3407
Xin Deng (iD) https://orcid.org/0000-0003-1580-0089

Reviewer #1 (Public review): https://doi.org/10.7554/eLife.96290.3.sa1
Reviewer #2 (Public review): https://doi.org/10.7554/eLife.96290.3.sa2
Author response https://doi.org/10.7554/eLife.96290.3.sa3

# Additional files

### Supplementary files

Supplementary file 1. Restriction–modification systems predicted in *P. syringae*.

Supplementary file 2. Modified genes conserved in three *P. syringae* strains.

Supplementary file 3. Differentially expressed genes (DEGs) between *Psph* WT and Δ*hsdMSR*.

Supplementary file 4. Differentially expressed genes (DEGs) carrying HsdMSR motif in their putative promoter regions.

Supplementary file 5. Genes with changed translational efficiency (TE) between *Psph* WT and Δ*hsdMSR*.

Supplementary file 6. Strains, plasmids, and primers.

MDAR checklist

### Data availability

The Ribo-seq, RNA-seq data, and SMRT-seq data were uploaded to the National Center for Biotechnology Information SRA database as part of BioProject PRJNA1055550 and PRJNA1123379, respectively.

The following datasets were generated:

| Author(s) | Year | Dataset title | Dataset URL | Database and Identifier |
| --- | --- | --- | --- | --- |
| Huang J, Chen F, Lu B, Sun Y, Li Y, Hua C, Deng X | 2024 | DNA Methylome Regulates Virulence and Metabolism in *Pseudomonas syringae* | https://www.ncbi.nlm.nih.gov/bioproject/?term=PRJNA1055550 | NCBI BioProject, PRJNA1055550 |
| Huang J, Chen F, Lu B, Sun Y, Li Y, Hua C, Deng X | 2024 | DNA Methylome Regulates Virulence and Metabolism in Pseudomonas syringae | https://www.ncbi.nlm.nih.gov/bioproject/?term=PRJNA1123379 | NCBI BioProject, PRJNA1123379 |

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
