## [Editor Report · eLife Assessment]

This **valuable** study presents findings on DNA methylation as an efficient epigenetic transcriptional regulating strategy in bacteria. The authors utilized single-molecule real-time sequencing to profile the DNA methylation landscape across three model pathovars of *Pseudomonas syringae*, identifying significant epigenetic mechanisms through the Type-I restriction-modification system, which includes a conserved sequence motif associated with N6-methyladenine. The evidence presented is **solid** and the study provides novel insights into the epigenetic mechanisms of *P. syringae*, expanding the understanding of bacterial pathogenicity and adaptation.

---

## [Referee Report · Reviewer #1 (Public review)]

Summary:

In this work, Huang et al used SMRT sequencing to identify methylated nucleotides (6mA, 4mC, and 5mC) in Pseudomonas syringae genome. They show that the most abundant modification is 6mA and they identify the enzymes required for this modification as when they mutate HsdMSR they observe a decrease of 6mA. Interestingly, the mutant also displays phenotypes of change in pathogenicity, biofilm formation, and translation activity due to a change in gene expression likely linked to the loss of 6mA.

Overall, the paper represents an interesting set of new data that can bring forward the field of DNA modification in bacteria.

Comments on revisions:

Thank you for the additional work. The authors have now addressed all my concerns.

---

## [Referee Report · Reviewer #2 (Public review)]

In the present manuscript, Huang et.al. revealed the significant roles of the DNA methylome in regulating virulence and metabolism within Pseudomonas syringae, with a particular focus on the HsdMSR system in this model strain. The authors used SMRT-seq to profile the DNA methylation patterns (6mA, 5mC, and 4mC) in three P. syringae strains (Psph, Pss, and Psa) and displayed the conservation among them. They further identified the type I restriction-modification system (HsdMSR) in P. syringae, including its specific motif sequence. The HsdMAR participated in the process of metabolism and virulence (T3SS & Biofilm formation), as demonstrated through RNA-seq analyses. Additionally, the authors revealed the mechanisms of the transcriptional regulation by 6mA. Strictly from the point of view of the interest of the question and the work carried out, this is a worthy and timely study that uses third-generation sequencing technology to characterize the DNA methylation in P. syringae. The experimental approaches were solid, and the results obtained were interesting and provided new information on how epigenetics influences the transcription in P. syringae. The conclusions of this paper are mostly well supported by data.

Comments on revisions:

The authors have successfully addressed all the comments from the reviewers in their revised manuscript.

---

## [Author Response]

The following is the authors’ response to the original reviews.

**Public Reviews:**

**Reviewer #1 (Public Review):**
Summary:In this work, Huang et al used SMRT sequencing to identify methylated nucleotides (6mA, 4mC, and 5mC) in Pseudomonas syringae genome. They show that the most abundant modification is 6mA and they identify the enzymes required for this modification as when they mutate HsdMSR they observe a decrease of 6mA. Interestingly, the mutant also displays phenotypes of change in pathogenicity, biofilm formation, and translation activity due to a change in gene expression likely linked to the loss of 6mA. Overall, the paper represents an interesting set of new data that can bring forward the field of DNA modification in bacteria.

Thank you for your valuable feedback on our paper exploring the impact of 6mA modification in P. syringae.

Major Concerns:Most of the authors' data concern Psph pathovar. I am not sure that the authors' conclusions are supported by the two other pathovars they used in the initial 2 figures. If the authors want to broaden their conclusions to Pseudomonas syringe and not restrict it to Psph, the authors should have stronger methylation data using replicates. Additionally, they should discuss why Pss is so different than Pst and Psph. Could they do a blot to confirm it is really the case and not a sequencing artifact? Is the change of methylation during bacterial growth conserved between the pathovar? The authors should obtain mutants in the other pathovar to see if they have the same phenotype. The authors have a nice set of data concerning Psph but the broadening of the results to other pathovar requires further investigation.

We appreciate the reviewer’s insightful comments. While the majority of our data focuses on the Psph, we recognize the importance of validating these findings in Pss and Pst. To this end, we have performed additional experiments using dot blot and mutant construction to enhance our conclusions in other pathovars.

We agree that we should discuss why Pss is different from Psph and Pst. We performed a dot blot assay using genome DNA in Pss and Pst, presented in Figure S5A. Meanwhile, we compared the 6mA modification level of Pss and Pst in different growth phases. As shown in Figure S5A, the change of methylation during bacterial growth is conserved in Pst. The change was not obvious in Pss, which might be due to the lack of a type I R-M system.

“In accordance with previous studies showing that growth conditions affect the bacterial methylation status, we applied dot blot experiments using the same amount of DNA (1 μg) from these three P. syringae strains to detect the 6mA levels during both logarithmic and stationary phases. The results revealed that 6mA levels in the stationary phase were much higher compared to the logarithmic phase in Psph and Pst, but no significant change in Pss. Additionally, we found that during the stationary phase, 6mA methylation levels in Psph and Pst were higher than those in Pss. These findings were consistent with the MTases predication on these three strains, since Pss does not harbor any type I R-M systems, which are important for 6mA medication in bacteria.”

Please see Figure S5A and Lines 220-228 in the revised manuscript.

We also tried to construct an HsdM mutant in Pst to explore whether the influence of 6mA methylation was conserved in P. syringae, but it failed after multiple attempts. We did not construct a Pss mutant because no type I R-M system was predicted, and few methylation sites were identified via SMRT-seq in this strain. Therefore, we overexpressed HsdM in Pst instead. We have performed additional experiments in WT and the HsdM overexpression strains, including dot blot and growth curve assays.

Please see Figures S5B-C and Lines228-232 in the revised manuscript.

The authors should include proper statistical analysis of their data. A lot of terms are descriptive but not supported by a deeper analysis to sustain the conclusions. For example, in Figure 4E, we do not know if the overlap is significant or not. Are DEGs more overlapping to 6mA sites than non-DEGs? Here is a non-exhaustive list of terms that need to be supported by statistics: different level (L145), greater conservation (L162), significant conservation (L165), considerable similarity (L175), credible motifs (L189), Less strong (L277) and several "lower" and "higher" throughout the text.

Thank you for the insightful feedback. We have made the following revisions in the manuscript to ensure that the terms are more precise and do not require statistical significance testing.

(1) Statistical analysis: We have added statistical tests for the overlap between DEGs and 6mA sites in Figure 4E. We performed the Fisher test, and we found the overlap was not significant (p> 0.05). DEGs and non-DEGs were both non-significant overlapped 6mA sites. Please see Figure 4E and Lines 261-262.

“Less strong” was used to indicate the influence of HsdM on biofilm in Figure 5D. All Figures with “*” labels were analyzed using students' two-tailed t-tests with a significant change (p < 0.05).

(2) Revised wording: For terms used to describe comparisons, we have revised the wording to be clearer and ensure that the terminology used did not imply the need for statistical significance testing when not required. For example:

“Different level” has been removed. Please see Lines 143-144.

“Greater conservation” has been revised to “more conserved functional terms”. Please see Lines 161-162.

“Significant conservation” has been revised to “notable conservation”. Please see Line 165.

“Credible motifs” has been revised to “identified motifs”. Please see Line 186.

The authors performed SMRT sequencing of the delta hsdMSR showing a reduction of 6mA. Could they include a description of their results similar to Figures 1-2. How reduced is the 6mA level? Is it everywhere in the genome? Does it affect other methylation marks? This analysis would strengthen their conclusions.

Yes, we agree. We have provided additional analysis and descriptions to strengthen the conclusions regarding these valuable comments. We determined three methylation sites in the HsdMSR mutant strain and compared the overlapped genes within these modification patterns. Specifically, we focused on the 6mA sites in Psph WT, HsdMSR mutant, and HsdM motif CAGCN_(6)_CTC. As expected, we found almost all of the reduction 6mA sites in the ΔhsdMSR were from motif CAGCN_(6)_CTC. We also noticed that 5mC and 4mC sites in the mutant were relatively similar to that in WT, and the slight difference might be caused by sequencing errors. Overall, we propose that HsdMSR only catalyze the 6mA located on the motif CAGCN_(6)_CTC, but does not affect other 6mA sites and other modification types.

Please see Figures S4D-E and Lines 212-218 in the revised manuscript.

In Figure 6E to conclude that methylation is required on both strands, the authors are missing the control CAGCN6CGC construct otherwise the effect could be linked to the A on the complementary strand.

Thank you for your valuable suggestions. We have provided the control result on the complementary strand. Please see Figure 6C. The new result evidences the conclusion that 6mA methylation regulates gene transcription based on methylation on both strands.

Please see Figure 6C and Lines 329-330 in the revised manuscript.

**Reviewer #2 (Public Review):**
In the present manuscript, Huang et.al. revealed the significant roles of the DNA methylome in regulating virulence and metabolism within Pseudomonas syringae, with a particular focus on the HsdMSR system in this model strain. The authors used SMRT-seq to profile the DNA methylation patterns (6mA, 5mC, and 4mC) in three P. syringae strains (Psph, Pss, and Psa) and displayed the conservation among them. They further identified the type I restriction-modification system (HsdMSR) in P. syringae, including its specific motif sequence. The HsdMAR participated in the process of metabolism and virulence (T3SS & Biofilm formation), as demonstrated through RNA-seq analyses. Additionally, the authors revealed the mechanisms of the transcriptional regulation by 6mA. Strictly from the point of view of the interest of the question and the work carried out, this is a worthy and timely study that uses third-generation sequencing technology to characterize the DNA methylation in P. syringae. The experimental approaches were solid, and the results obtained were interesting and provided new information on how epigenetics influences the transcription in P. syringae. The conclusions of this paper are mostly well supported by data, but some aspects of data analysis and discussion need to be clarified and extended.

Thank you for your positive feedback and recognition of the importance of our study. We appreciate the suggestions for further clarification and extension of some aspects of data analysis and discussion. We added further analysis of the SMRT-seq result of the ΔhsdMSR and overexpressed HsdM in Pst to provide more information on conservation. We added these contents to the discussion in the revised manuscript. Please see Figure 6C and Figure S5.

**Reviewer #3 (Public Review):**
Summary:The article by Huang et.al. presents an in-depth study on the role of DNA methylation in regulating virulence and metabolism in Pseudomonas syringae, a model phytopathogenic bacterium. This comprehensive research utilized single-molecule real-time (SMRT) sequencing to profile the DNA methylation landscape across three model pathovars of P. syringae, identifying significant epigenetic mechanisms through the Type-I restriction-modification system (HsdMSR), which includes a conserved sequence motif associated with N6-methyladenine (6mA). The study provides novel insights into the epigenetic mechanisms of P. syringae, expanding the understanding of bacterial pathogenicity and adaptation. The use of SMRT sequencing for methylome profiling, coupled with transcriptomic analysis and in vivo validation, establishes a robust evidence base for the findingsStrengths:The results are presented clearly, with well-organized figures and tables that effectively illustrate the study's findings.Weaknesses:It would be helpful to add more details, especially in the methods, which make it easy to evaluate and enhance the manuscript's reproducibility.

Thank you for the positive evaluation of our study, as well as the constructive feedback provided. We have added more details in methods for RNA-seq analysis and Ribo-seq analysis. Please see Lines 484-515.

“Briefly, bacteria were cultured to an OD_600_ of 0.4, at which point chloramphenicol was added to a final concentration of 100 µg/mL for 2 minutes. Cells were then pelleted and washed with pre-chilled lysis buffer [25 mM Tris-HCl, pH 8.0; 25 mM NH4Cl; 10 mM MgOAc; 0.8% Triton X-100; 100 U/mL RNase-free DNase I; 0.3 U/mL Superase-In; 1.55 mM chloramphenicol; and 17 mM GMPPNP]. The pellet was resuspended in lysis buffer, followed by three freeze-thaw cycles using liquid nitrogen. Sodium deoxycholate was then added to a final concentration of 0.3% before centrifugation. The resulting supernatant was adjusted to 25 A260 units and mixed with 2 mL of 500 mM CaCl_2_ and 12 µL MNase, making up a total volume of 200 µL. After the digestion, the reaction was quenched with 2.5 mL of 500 mM EGTA. Monosomes were isolated using Sephacryl S400 MicroSpin columns, and RNA was purified using the miRNeasy Mini Kit (Qiagen). rRNA was removed using the NEBNext rRNA Depletion Kit, and the final library was constructed with the NEBNext Small RNA Library Prep Kit. For each sample, ribosome footprint reads were mapped to the Psph 1448A reference genome, and the translational efficiency was calculated by dividing the normalized Ribo-seq counts by the normalized RNA counts. Two biological replicates were performed for all experiments.”

**Recommendations For The Authors:**

**Reviewer #1 (Recommendations For The Authors):**
I would recommend the authors limit their manuscript to Psph pathovar and include statistical analysis supporting their conclusions.

Thank you for your suggestion.

Minor• L104: "significantly" please add a p-value and explain the analysis.

Sorry for the confusion. We have added the p-value and explained the analysis in the method section. The p-value used for SMRT-seq was the modification quality value (QV) score, which were used to call the modified bases A (QV=50) and C (QV=100). Please see Lines 452-454.

• Figures 1B, D, F, and Figure 2A: make the Venn diagram to scale

Yes, we have revised.

• L110-111: missing p-value to say that the authors observe a bigger overlap in Pst than Psph as they observed more modified sites in Pst

Sorry for the confusion. We said it had a bigger overlap in Pst because the number 17.7 was bigger than the number of 15 in Psph. To avoid misunderstanding, we revised the wording to “more genes equipped with all three modification types were detected in Pst than Psph”. Please see Lines 110-111.

• L112: missing description of their Pss analysis (IDP, sites...)

We have added the information for Pss in the revised manuscript.

“Additionally, the methylome atlas of Pss revealed a lower incidence of methylation than those of Psph and Pst, particularly in terms of 6mA modifications, which were only seen in 457 significant 6mA occurrences under the same threshold (IPD > 1.5) and a total of 2,853 and 1,438 methylation sites were detected as 5mC and 4mC, respectively”. Please see Lines 114-116.

• L118: "modification" to "modified "

We have revised. Please see Line 119.

• L120: "modification sites" to "modified nucleotides"

We have revised. Please see Line 121.

• L142: correct the title "Methylated genes revealed highly functional conservation among three P. syringae strains" maybe to "Methylated genes are functionally conserved among ..."

We have revised. Please see Line 142.

• Figure 2C: not easy to read and interpret

Sorry for the confusion. Figure 2C revealed the significantly enriched functional pathways in GO and KEGG databases among three P. syringae strains. The specific names of each pathway were listed on the left, and each column with dots indicated the number of genes within one kind of methylation in one of three P. syringae strains. The larger the size, the bigger the number.

We have revised the legend of Figure 2C. Please see Lines 575-579.

“The dot plot revealed the significantly enriched functional pathways in GO and KEGG databases among three P. syringae strains. The specific names of each pathway were listed on the left, and each column with dots indicated the number of genes within one kind of methylation in one of three P. syringae strains. The size of the dots indicates the number of related genes.”

• Figure 6B-C: what is the difference between B 24h and C?

Figure 6B revealed the expression difference between WT and mutant during 24 hours. Figure 6C only showed a time point in 24 hours. To avoid repetition, we have removed Figure 6C.

• Figure 6C-D: if the same maybe remove Figure 2C

We have removed Figure 6D.

**Reviewer #2 (Recommendations For The Authors):**
The manuscript could be improved by addressing the following concerns:(1) In line 146: How to understand the percentage conserved in "more than two of the strains"?

Sorry for the confusion, we planned to indicate the pattern that conserved in two strains and three strains. We have revised it to: “Notable, about 25% to 45% of methylated genes were conserved in two and three strains”. Please see Line 145.

(2) In line 178: Five conserved sequence motifs should be replaced by "Six conserved sequence motifs".

We have revised. Please see Line 176.

(3) In Figure 2B, specify the C1, C2 and C3. "m6A" should be replaced by "6mA".

Yes, we have revised.

(4) In Figure S2, "m6A" should be replaced by "6mA".

Yes, we have revised.

(5) In line 212, please add references for the previous studies showing that growth conditions affect bacterial methylation status.

Thank you for your suggestion. We have added the relevant references (Gonzalez and Collier, 2013), (Krebes et al., 2014), (Sanchez-Romero and Casadesus, 2020).

(6) In line 217, "illustrate" should be "illustrated".

Yes, we have revised. Please see Line 210.

(7) There are some genes colored in grey, revealing bigger differences between the two strains than those related to ribosomal protein, T3SS, and alginate synthesis in Fig. 4A. Do they have important functional roles as well?

Thank you for your suggestion. A total of 116 genes with bigger differences (|Log_2_FC| > 2) except for genes related to ribosomal protein, T3SS, and alginate synthesis. Among these genes, 31 were annotated as hypothetical proteins and 4 as transcription factors with unknown functions, and the remaining genes mostly encoded metabolism-related enzymes. These enzymes might have effects on growth defects in ΔhsdMSR. We added this information in the revised manuscript. Please see Line 249-254.

(8) The authors should discuss what will be the potential signals or factors that can regulate the activity of HsdMSR. In other words, what can decide the extent of methylation through activating or suppressing the expression of HsdMSR?

Thank you for your valuable suggestion. We have added this part in the discussion part. Please see Lines 404-415.

“Apart from the established roles of 6mA and HsdMSR in P. syringae, certain signals or factors may influence HsdMSR expression. For instance, we confirmed that the growth phase affects methylation levels in P. syringae. Previous studies have shown that increased temperatures can reduce methylation levels, as observed in *P. aeruginosa* PAO1(Doberenz et al., 2017). These findings suggest that HsdMSR expression may be responsive to both intrinsic cellular states and extrinsic environmental conditions. To further explore potential upstream TFs regulating the expression of HsdMSR, we searched for TF-binding sites in the HsdMSR promoter using our own databases (Fan et al., 2020; Shao et al., 2021; Sun et al., 2024). As a result, we found three candidate TFs (PSPPH_0061, PSPPH_3268, and PSPPH_3504) that might directly bind and regulate HsdMSR expression. Future studies on these TFs and their interactions with the HsdMSR promoter would help clarify the regulatory network governing HsdMSR activity.”

**Reviewer #3 (Recommendations For The Authors):**
(1) Some figures contain dense information, which may be overwhelming for readers. Streamlining the legend for Figure 1 and resizing the Venn diagrams within it could enhance clarity and visual appeal.

Thank you for your suggestion. We have scaled all the Venn plots in the revised version.

(2) Incorporating a discussion about the role of the restriction-modification (RM) system in bacterial defense against phage infection into the discussion section could enrich the manuscript's context and relevance.

Thank you for your valuable suggestion. We have added this part in the Discussion part. Please see Lines 416-427.

“RM systems are known for their intrinsic role as innate immune systems in anti-phage infection, and present in around 90% of bacterial genomes(Oliveira et al., 2014). RM systems protect bacteria self by recognizing and degrading foreign phage DNA via methylation-specific site and restriction endonucleases (REases) (Loenen et al., 2014). In addition, other phage-resistance systems are similar to RM systems but carry extra genes. One is called the phage growth limitation (Pgl) system, which modifies and cleaves phage DNA. However, the Pgl only modifies the phage DNA in the first infection cycle, and cleaves phage DNA in the subsequent infections, which gives a warn to the neighboring cells(Hampton et al., 2020; Hoskisson et al., 2015). To counteract RM and RM-like systems, phages have evolved strategies, including unusual modifications such as hydroxymethylation, glycosylation, and glucosylation. They can also encode their own MTases to protect their DNA or employ strategies to evade restriction systems and other anti-RM defenses.(Iida et al., 1987; Murphy et al., 2013; Vasu and Nagaraja, 2013).

(3) In line 152: What is the importance of the mentioned example of Cro/CI family TF?

Thank you for your comments. The Cro/CI are important TFs present in phages. The interaction between Cro and CI affects bacteria immunity status in Enterohemorrhagic *Escherichia coli* (EHEC) strains(Jin et al., 2022). RM systems are known as a kind of phage-defense system, and hence we mentioned it here. We have added this description in the revised manuscript. Please see Lines 152-154.

Reference:

(1) Doberenz, S., Eckweiler, D., Reichert, O., Jensen, V., Bunk, B., Sproer, C., Kordes, A., Frangipani, E., Luong, K., Korlach, J., et al. (2017). Identification of a *Pseudomonas aeruginosa* PAO1 DNA Methyltransferase, Its Targets, and Physiological Roles. mBio 8. 10.1128/mBio.02312-16.

(2) Fan, L., Wang, T., Hua, C., Sun, W., Li, X., Grunwald, L., Liu, J., Wu, N., Shao, X., Yin, Y., et al. (2020). A compendium of DNA-binding specificities of transcription factors in Pseudomonas syringae. Nat Commun 11, 4947. 10.1038/s41467-020-18744-7.

(3) Gonzalez, D., and Collier, J. (2013). DNA methylation by CcrM activates the transcription of two genes required for the division of Caulobacter crescentus. Mol Microbiol 88, 203-218. 10.1111/mmi.12180.

(4) Hampton, H.G., Watson, B.N., and Fineran, P.C. (2020). The arms race between bacteria and their phage foes. Nature 577, 327-336.

(5) Hoskisson, P.A., Sumby, P., and Smith, M.C. (2015). The phage growth limitation system in Streptomyces coelicolor A (3) 2 is a toxin/antitoxin system, comprising enzymes with DNA methyltransferase, protein kinase and ATPase activity. Virology 477, 100-109.

(6) Iida, S., Streiff, M.B., Bickle, T.A., and Arber, W. (1987). Two DNA antirestriction systems of bacteriophage P1, darA, and darB: characterization of darA− phages. Virology 157, 156-166.

(7) Jin, M., Chen, J., Zhao, X., Hu, G., Wang, H., Liu, Z., and Chen, W.-H. (2022). An engineered λ phage enables enhanced and strain-specific killing of enterohemorrhagic *Escherichia coli*. Microbiology Spectrum 10, e01271-01222.

(8) Krebes, J., Morgan, R.D., Bunk, B., Sproer, C., Luong, K., Parusel, R., Anton, B.P., Konig, C., Josenhans, C., Overmann, J., et al. (2014). The complex methylome of the human gastric pathogen Helicobacter pylori. Nucleic Acids Res 42, 2415-2432. 10.1093/nar/gkt1201.

(9) Loenen, W.A., Dryden, D.T., Raleigh, E.A., Wilson, G.G., and Murray, N.E. (2014). Highlights of the DNA cutters: a short history of the restriction enzymes. Nucleic Acids Res 42, 3-19.

(10) Murphy, J., Mahony, J., Ainsworth, S., Nauta, A., and van Sinderen, D. (2013). Bacteriophage orphan DNA methyltransferases: insights from their bacterial origin, function, and occurrence. Appl Environ Microb 79, 7547-7555.

(11) Oliveira, P.H., Touchon, M., and Rocha, E.P. (2014). The interplay of restriction-modification systems with mobile genetic elements and their prokaryotic hosts. Nucleic Acids Res 42, 10618-10631.

(12) Sanchez-Romero, M.A., and Casadesus, J. (2020). The bacterial epigenome. Nature reviews. Microbiology 18, 7-20. 10.1038/s41579-019-0286-2.

(13) Shao, X., Tan, M., Xie, Y., Yao, C., Wang, T., Huang, H., Zhang, Y., Ding, Y., Liu, J., Han, L., et al. (2021). Integrated regulatory network in Pseudomonas syringae reveals dynamics of virulence. Cell Rep 34, 108920. 10.1016/j.celrep.2021.108920.

(14) Sun, Y., Li, J., Huang, J., Li, S., Li, Y., Lu, B., and Deng, X. (2024). Architecture of genome-wide transcriptional regulatory network reveals dynamic functions and evolutionary trajectories in Pseudomonas syringae. bioRxiv, 2024.2001. 2018.576191.

(15) Vasu, K., and Nagaraja, V. (2013). Diverse functions of restriction-modification systems in addition to cellular defense. Microbiol Mol Biol Rev 77, 53-72. 10.1128/MMBR.00044-12.